# Respiratory mucosal immune memory to SARS-CoV-2 after infection and vaccination

Elena Mitsi [1,2,8] ✉, Mariana O. Diniz[3,8], Jesús Reiné[1,2], Andrea M. Collins[2], Ryan E. Robinson[2], Angela Hyder-Wright[2], Madlen Farrar[2], Konstantinos Liatsikos [2], Josh Hamilton [2], Onyia Onyema[2], Britta C. Urban[1,2], Carla Solórzano[1,2], Sandra Belij-Rammerstorfer[1], Emma Sheehan[1], Teresa Lambe [1,4], Simon J. Draper [5], Daniela Weiskopf[6], Alessandro Sette [6,7], Mala K. Maini [3,9] & Daniela M. Ferreira [1,2,9] ✉

Respiratory mucosal immunity induced by vaccination is vital for protection from coronavirus infection in animal models. In humans, the capacity of peripheral vaccination to generate sustained immunity in the lung mucosa, and how this is influenced by prior SARS-CoV-2 infection, is unknown. Here we show using bronchoalveolar lavage samples that donors with history of both infection and vaccination have more airway mucosal SARS-CoV-2 antibodies and memory B cells than those only vaccinated. Infection also induces populations of airway spike-specific memory CD4+ and CD8+ T cells that are not expanded by vaccination alone. Airway mucosal T cells induced by infection have a distinct hierarchy of antigen specificity compared to the periphery. Spike-specific T cells persist in the lung mucosa for 7 months after the last immunising event. Thus, peripheral vaccination alone does not appear to induce durable lung mucosal immunity against SARS-CoV-2, supporting an argument for the need for vaccines targeting the airways.

Respiratory mucosal surfaces are the primary site of interaction between SARS-CoV-2 and the immune system. Mucosal antibodies and tissue-resident memory T (TRM) and B cells provide frontline early responses, contributing to protection against the establishment of viral infection following previous viral exposure or vaccination[1–3]. Animal studies of influenza virus infection have shown that development of antigen-specific resident memory B cells in the lung produces local IgG and IgA with enhanced cross-recognition of variants[4] and correlates with protection against reinfection in mice[5,6]. Additionally, studies of respiratory viral infections in animals and human-controlled challenge have highlighted the essential role of local tissue-memory T cells in promoting immunity against influenza and respiratory syncytial virus (RSV) mediated, at least in part, by rapid IFN-γ production[7–12]. Interestingly, in a murine model of SARS-1 and MERS coronavirus infection, protection was attributed to the induction of CD4+ T cells in the airway[13].

The difficulty accessing human mucosal sites, particularly the lower airways, and the low cell yield, have hindered the study of local immunity to respiratory pathogens. Most human studies have assessed antibody and T-cell responses to SARS-CoV-2 in blood, which is often not reflective of the responses in the airways. We and others have demonstrated the presence of pre-existing T cells that recognise SARS-CoV-2 in the lower airways or oropharyngeal lymphoid tissue of unexposed individuals respectively[14,15], likely induced by seasonal

[1]Oxford Vaccine Group, Department of Paediatrics, University of Oxford, Oxford, UK. [2]Department of Clinical Science, Liverpool School of Tropical Medicine, Liverpool, UK. [3]Division of Infection and Immunity and Institute of Immunity and Transplantation, UCL, London, UK. [4]Chinese Academy of Medical Science (CAMS) Oxford Institute (COI), University of Oxford, Oxford, UK. [5]Department of Biochemistry, University of Oxford, Oxford, UK. [6]Center for Infectious Disease and Vaccine Research, La Jolla Institute for Immunology (LJI), La Jolla, USA. [7]Department of Medicine, Division of Infectious Diseases and Global Public Health, University of California, San Diego, La Jolla, USA. [8]These authors contributed equally: Elena Mitsi, Mariana O. Diniz. [9]These authors jointly supervised this work: Mala K. Maini, Daniela M. Ferreira. ✉e-mail: elena.mitsi@paediatrics.ox.ac.uk; daniela.ferreira@paediatrics.ox.ac.uk

coronavirus infections. Presence of SARS-CoV-2 specific T cells were also reported in human nasal[16], lung mucosa and lung-associated lymph nodes following SARS-CoV-2 infection[17–19]. Furthermore, increased numbers of global CD4+ and CD8+ in the airways of SARS-CoV-2-infected patients were associated with reduced disease severity[20,21]. It has also been reported that spike-specific memory B cells were enriched in the lungs and associated lymph nodes of convalescent organ donors[19] and that SARS-CoV-2-binding IgA antibodies are produced more rapidly than IgG and can be detected in the serum and saliva of COVID-19 patients up to 40 days post onset of symptoms[22–25].

Recent animal work with different SARS-CoV-2 vaccine formulations showed the need for mucosal immunisation to generate resident virus-specific B and T cells in the lungs and confirmed the importance of localised mucosal immunity in control of infection[17,26–28]. Human studies that described the effect of peripheral SARS-CoV-2 vaccines on the respiratory mucosa are conflicting. While nasal and salivary IgA responses[29], as well as CD4+ and CD8+ $T_{RM}$ were detected in the nasal mucosa[30] of vaccinated individuals without history of SARS-CoV-2 infection[29], other studies reported minimal or lack of humoral and T cell responses in nasal and lung mucosa following peripheral vaccination only[17,31]. However, such responses were detected in convalescent donors or after breakthrough infection[17,19,31,32].

Further studies are needed to better characterise immune responses in the airways after infection and/or vaccination and dissect out the influence of hybrid immunity, vaccine type, disease severity, and particularly time since vaccination or infection to address persistence of mucosal immunity. Using bronchoalveolar lavage (BAL) samples collected before the onset of the COVID-19 pandemic, we have previously demonstrated that SARS-CoV-2-cross-reactive T cells can reside in human airways[14]. Here, we tested BAL samples and paired blood from vaccinated donors with or without SARS-CoV-2 infection and pre-pandemic control samples. We examined the presence of peripheral and mucosal antibodies and virus-specific B and T cell responses. Spike- and RBD-specific B cells were only detected in the airways of infected vaccinated individuals. Similarly, virus-specific CD4+ and CD8+ T cells were more abundant in this group compared to uninfected vaccinated individuals. A better understanding of the breadth and longevity of adaptive immunity to SARS-CoV-2 in the airways will allow us to harness protective mucosal immunity to inform next generation SARS-CoV-2 vaccines with potential to block infection and population transmission.

## Results
### Characteristics of study groups
To assess humoral and cellular immune responses in the lung mucosa and blood following SARS-CoV-2 vaccination and hybrid immunity, we collected bronchoalveolar lavage (BAL) fluid and paired blood samples from 9 vaccinees with no history or evidence of SARS-CoV-2 infection (uninfected vaccinated group) and 22 vaccinees who had serologically confirmed asymptomatic infection ($n = 5$) or experienced symptomatic infection prior to receiving SARS-CoV-2 vaccination ($n = 13$) or after receiving their booster vaccine ($n = 4$; Fig. 1A and Supplemental Fig. 1). Vaccinated individuals with asymptomatic and symptomatic infection were combined in one group of infected vaccinated subjects, due to the lack of obvious difference in immune responses to SARS-CoV-2 antigens. Vaccinated individuals received either two or three doses of mRNA, adenoviral vector vaccine or a combination of them (Table 1). We also included a pre-pandemic group ($n = 11$) of unexposed, unvaccinated individuals as controls (Fig. 1A). Table 1 summarises the demographic characteristics of the three study groups and the time of sample collection in relation to infection or last vaccination.

### Airway antibody responses following vaccination with or without infection
Levels of circulating and mucosal antibodies against spike (S), receptor binding domain (RBD) and nucleocapsid (N) protein were measured in serum and BAL samples in all three study groups. Antibody responses to N protein (non-vaccine protein) were used to confirm the absence of past infection when classifying the groups. The limit of sensitivity (LOS) was set as median + 2 x standard deviation (SD) of the results in unexposed (pre-pandemic) donors. As expected, anti-N IgG was below or near the LOS in the uninfected vaccinated group, whereas in the infected vaccinated group, it was detected in all individuals (22/22) in serum and in 68% (15/22) in the BAL fluid (Suppl. Fig. 2A, B).

SARS-CoV-2 vaccination elicited systemic IgG responses to both S and RBD protein, with levels being more pronounced in the infected vaccinated group (6.6-fold and 3.5-fold median increase of anti-S and anti-RBD IgG compared to uninfected vaccinated group, respectively) ($p = 0.027$ and $p > 0.05$, respectively) (Fig. 1B, C). Such systemic antibody differences as a result of hybrid immunity have been extensively demonstrated in large cohort vaccination studies[33,34]. High levels of anti-S and -RBD IgG were also detected in the BAL fluid of SARS-CoV-2 vaccinees. Importantly, anti-S and anti-RBD IgG levels in the BAL were also significantly elevated in the infected vaccinated group compared to the uninfected vaccinated group (5-fold and 6.1-fold increase for S and RBD, respectively) ($p = 0.013$ and $p = 0.012$, respectively) (Fig. 1D, E).

As IgA plays a crucial role in the antiviral immune defence at mucosal surfaces[35], IgA responses against SARS-CoV-2 proteins were also assessed in BAL samples. In the uninfected vaccinated group, mucosal IgA levels against S, RBD and N did not differ from the control group. However, the infected vaccinated group had significantly greater mucosal anti-S IgA (3.3-fold increase from control, $p = 0.03$) and a trend to higher anti-RBD IgA (Fig. 1F, G), whereas the majority had non-detectable anti-N IgA levels in BAL (Suppl. Fig. 2C).

Vaccine-induced antibody responses to S protein demonstrated a strong correlation between serum and BAL for both IgG and IgA (Suppl. Fig. 2D, E).

### Detectable SARS-CoV-2 specific memory B cells in the lung mucosa of infected vaccinated donors
Memory B cells are critical for long-term humoral immunity. To identify SARS-CoV-2 specific memory B cells (MBCs), fluorescently labelled S, RBD and N proteins were used to assay PBMCs and lung leucocytes (Fig. 2A; see gating strategy, Suppl. Fig. 3A, B). As expected, and in line with the antibody responses, only vaccinees who had previously been exposed to viral nucleoprotein through infection had detectable N-specific MBCs in the blood (Fig. 2D). By contrast, both uninfected and infected vaccinated individuals had circulating S- and RBD-specific MBCs above the background staining threshold (set as median + 2 x SD of pre-pandemic levels) (Fig. 2B, C).

B cells are an underrepresented cell population in the respiratory mucosa; their presence in the human lung is usually associated with infection or chronic inflammation[36]. Although data on anti-viral B cell immunity in human respiratory mucosa are scarce, murine model studies of influenza infection demonstrated the generation of flu-specific memory B cells in the lung following influenza infection that were able to produce antibodies with enhanced potential to recognise viral variants[4–6]. In this study, the small number of B cells in the BAL samples only allowed the assessment of SARS-CoV-2 specific MBCs only in a subset of uninfected and infected vaccinated individuals. The frequencies of S-, RBD- and N-specific MBCs were significantly greater in the lung mucosa of infected vaccinated individuals compared to uninfected vaccinated, being undetectable in the latter group (median 4.2% vs 0.01% for S, 2.88% vs 0.01% for RBD and 1.69% vs 0.01% for N-specific responses ($p = 0.003$, $p = 0.006$ and $p = 0.001$, respectively) (Fig. 2E–G). Paired sample comparison of the frequencies of circulating

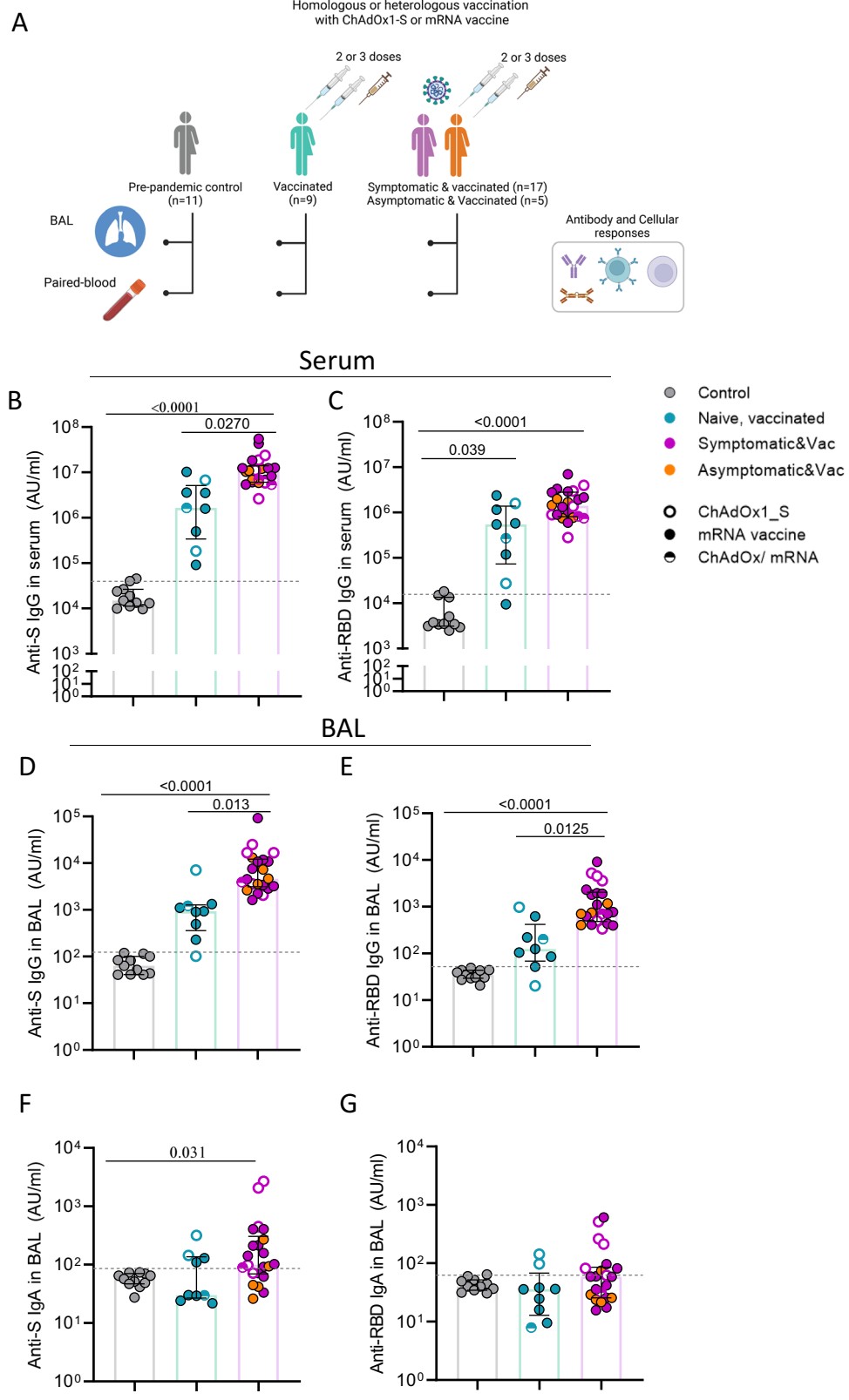

and mucosally detected anti-viral MBCs in the vaccinated groups revealed exclusive enrichment of S- and RBD-specific MBCs in the lung mucosa of infected vaccinated individuals. The median frequencies of S- and RBD-binding MBCs were 2.6-fold ($p = 0.013$) and 3.8-fold ($p = 0.062$) higher in the BAL compared to PBMC from the same donors (Fig. 2H). We detected that in the lung mucosa memory B cells

were mainly class-switched MBCs, whereas paired blood samples had a significantly increased proportion of unswitched MBCs cells (Fig. 2I).

Notably, the frequency of S-binding MBCs in the blood and BAL samples of infected vaccinated donors correlated with the serum and BAL anti-Spike IgG titres, respectively (Suppl. Fig. 2B, C). Similarly, the levels of RBD- and N-binding memory B cells detected in blood

**Fig. 1 | Systemic and lung mucosa antibody responses following vaccination and infection or vaccination alone. A** Schematic of study groups depicting SARS-CoV-2 vaccination status, sample collection per group and immunological parameters analysed. Pre-pandemic controls ($n = 11$), infection-naïve vaccinated individual (uninfected vaccinated group, $n = 9$) and vaccinated individuals with exposure to SARS-CoV-2 (infected vaccinated group, $n = 22$). Different colours used to depict convalescents with asymptomatic or symptomatic SARS-CoV-2 infection. **B–E** Levels of IgG against Spike (**B** and **D**) and RBD (**C** and **E**) in serum and bronchoalveolar lavage (BAL) fluid of control ($n = 11$), vaccinated ($n = 9$) and infected vaccinated donors ($n = 22$). **F, G** Levels of IgA against Spike (**F**) and RDB (**G**) in BAL fluid of control ($n = 11$), vaccinated ($n = 9$) and infected, vaccinated donors ($n = 22$). Antibody levels are expressed as arbitrary units (AU). The limit of assay sensitivity (LOS) per antigen is depicted with dotted black line. Homologous vaccination with ChAdOX1_S or mRNA vaccine is depicted with an open or close circle, respectively and heterologous vaccination with semi-full circle. Data are presented as median values and interquartile ranges (IQRs). Statistical differences were determined by Kruskal-Wallis test following correction for multiple comparisons. Adjusted $p$ values are shown. Source data are provided as a Source Data file.

associated positively with the relevant antibody titres in serum (Suppl. Fig. 3D, E).

### Induction of SARS-CoV-2 specific T cells responses in the lung mucosa after infection and vaccination but not vaccination alone

Circulating and tissue resident memory ($T_{RM}$) T cells are important in constraining viral spread and protect against severe disease when neutralising antibodies fail to confer sterilising immunity[37–39]. Moreover, we showed that T cells targeting the early expressed replication transcription complex (RTC: NSP7,12,13) are selectively associated with infection being aborted before detection by PCR or seroconversion and can be detected in pre-pandemic blood and BAL samples[14,40]. Therefore, we examined T cell responses in paired blood and BAL samples following vaccination alone or infection and vaccination.

The frequencies of circulating and lower airway CD4$^+$ and CD8$^+$ T cells were measured based on the expression of activation-induced markers (AIM assay) after stimulation with SARS-CoV-2 peptides (for full gating strategy see Suppl. Fig. 4) and were compared to pre-existing cross-reactive responses detectable using the same assays in cryopreserved pre-pandemic BAL samples. BAL samples were further divided by the expression of prototypic tissue residency markers (CD69/CD49a co-expression for CD4 and CD69/CD103 co-expression for CD8) into $T_{RM}$ and recirculating T cells. As reported by others[41–43], SARS-CoV-2 vaccination alone induced notable S-specific CD4$^+$ and CD8$^+$ T cell responses in the circulation when compared to pre-pandemic controls (Fig. 3B, C). In the infected vaccinated group, the frequency of circulating S-specific CD4$^+$ and CD8$^+$ T cells tended to be higher than the uninfected vaccinated group (2.7-fold and 5.5-fold increase, respectively). Despite the induction of T cell immunity systemically, vaccination alone did not elicit S-specific T cell responses that were significantly greater than those in pre-pandemic samples within the global (Fig. 3D, E) or $T_{RM}$ lung mucosa compartment (Fig. 3F, G).

As opposed to vaccination alone, BAL samples from those who were infected and vaccinated exhibited greater anti-Spike T cell responses than either the pre-pandemic or uninfected vaccinated group (Fig. 3D–G). Within the global T cell population, the frequency of S-specific CD4$^+$ and CD8$^+$ T cells increased by 4.6-fold and 18-fold higher in the infected vaccinated group compared to the uninfected vaccinated group ($p = 0.013$ and $p = 0.0009$, respectively) (Fig. 3D, E). A similar profile was observed in the $T_{RM}$ T cell compartment, with S-specific CD4$^+$ and CD8$^+$ T cell frequencies being 3.9-fold and 9.6-fold greater, respectively, in the infected vaccinated group compared to uninfected vaccinated group ($p = 0.015$ and $p = 0.004$, respectively) (Fig. 3F and G). In addition, within the global T cell population, the frequencies of S-specific CD4$^+$ and CD8$^+$ T cells were substantially higher in BAL than in paired blood from infected vaccinated individuals (median 3.72% vs 1.07% of S-specific CD4$^+$ T cells and median 1.84% vs 0.33% of S-specific CD8$^+$ T in BAL and paired blood of infected vaccinated individual, respectively) ($p < 0.0001$ and $p = 0.0002$, respectively) (Fig. 3H, I). In the uninfected vaccinated group, the frequency of S-specific CD4$^+$ but not CD8$^+$ T cells was slightly higher in BAL than PBMCs (median 0.81% vs 0.39% in BAL and paired blood, respectively).

We also examined T cell specificities to non-vaccine included SARS-CoV-2 structural proteins (N and membrane [M]) and non-structural proteins (NSP-7, NSP-12 and NSP-13 pool, representative of the core replication-transcription complex [RTC]) in blood and BAL (Suppl. Fig. 5A–D and Fig. 4). As expected, the frequencies of circulating N- and M-specific CD4$^+$ and CD8$^+$ T cells were only significantly higher than pre-pandemics in the infected vaccinated group, as those vaccinees would have been exposed to relevant antigens (Fig. 4A and B). In the case of RTC-specific T cells, their frequency did not differ amongst groups, as SARS-CoV-2 cross-reactive CD4$^+$ and CD8$^+$ T cell responses were detected systemically in 3 out 8 pre-pandemic controls, in line with previous studies[14,40,44]. In BAL samples, the frequency of the aforementioned T cell specificities was tested in a subset of pre-pandemic and infected vaccinated individuals based on cell number availability. Interestingly, the infected vaccinated group had, or tended to have, higher N- and M- and RCT-specific T cell responses within the global and TRM T cell compartment in BAL samples compared to levels detected in pre-pandemic controls (Fig. 4C–F). Overall, these SARS-CoV-2 specific CD4$^+$ and CD8$^+$ T cell responses were enriched in the lower airways compared to the periphery (Fig. 4G, H).

The hierarchy of SARS-CoV-2 antigen recognition by circulating and lower airway T cells of each distinct peptide pool (S, N, M and RTC) was analysed in a subset of 8 infected vaccinated individuals (Suppl. Figure 5A, B). The antigen recognition profile differed between systemic and airway localised T cells, and between T cell subsets. SARS-CoV-2 specific CD4$^+$ T cells were largely dominated

### Table 1 | Characteristics of participants

| Characteristics | Controls (n = 11) | Vaccinated (n = 9) | Infected and vaccinated (n = 22) |
|---|---|---|---|
| Age (year), median (IQR) | 21 (19–24) | 25 (23–41) | 35 (20–57) |
| Female, n (%) | 7 (64) | 5 (56) | 13 (59) |
| **Infection history** | | | |
| Time since symptomatic infection in days, median (min–max) | n/a | n/a | 266 (37–570) |
| NIH clinical score, median (IQR) | n/a | n/a | 3 (1-4) |
| Infection in relation to vaccination | n/a | n/a | Primary infection = 13<br>Breakthrough infection = 4<br>Unknown = 5 |
| **Vaccination history** | | | |
| Vaccine doses | n/a | 2 doses, n = 7<br>3 doses, n = 2 | 2 doses, n = 15<br>3 doses, n = 7 |
| Vaccine type | n/a | BNT162b2, n = 6<br>ChAdOx1_S, n = 2<br>ChAdOx1_S/ BNT162b2, n = 1 | BNT162b2, n = 15<br>Moderna, n = 1<br>ChAdOx1_S, n = 5<br>ChAdOx1_S/ BNT162b2, n = 1 |
| Time since 2nd or 3rd vaccine dose in days, median (min–max) | n/a | 130 (23–249) | 114 (23–392) |

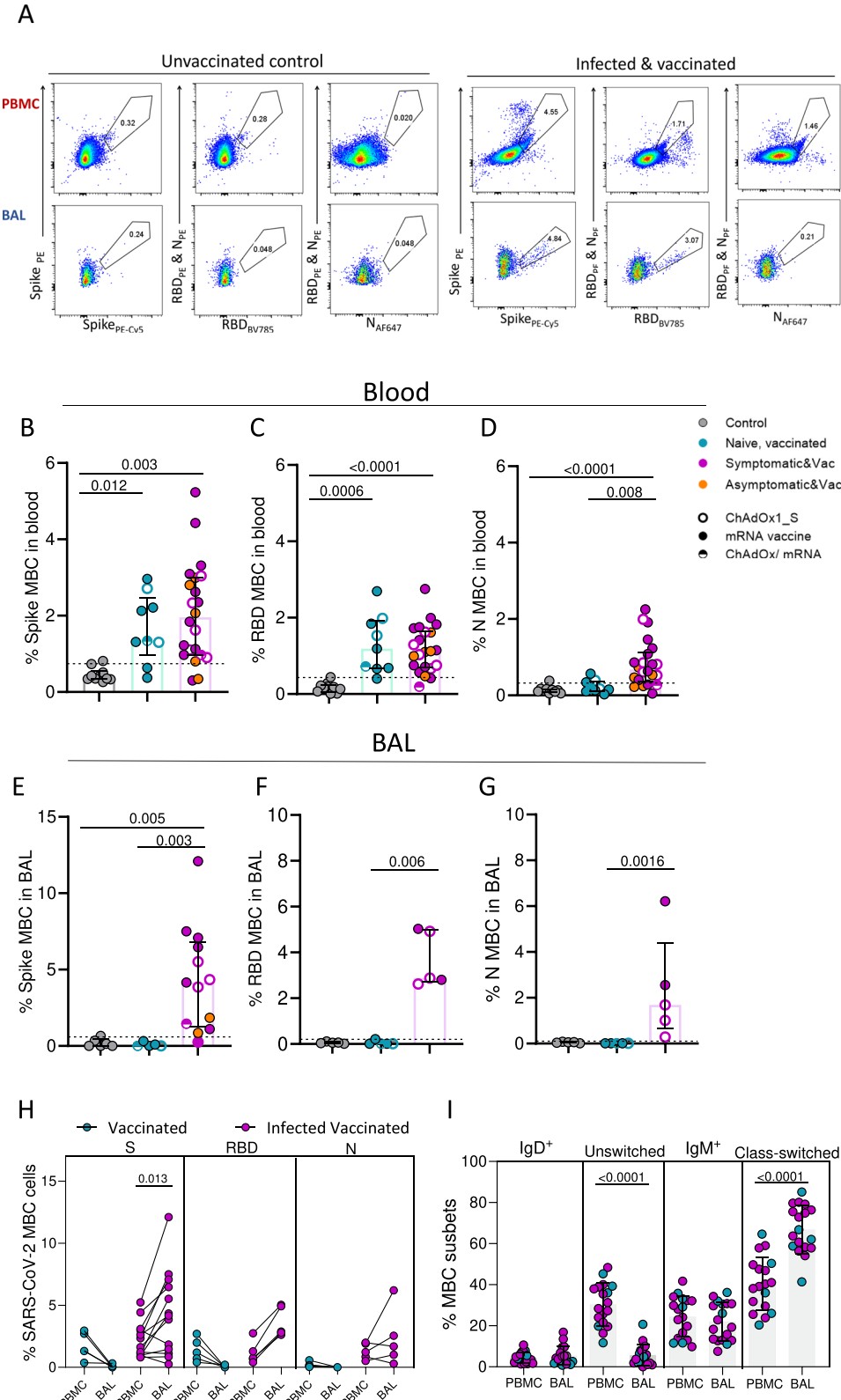

by S-specific CD4⁺ T cells in the periphery and lung mucosa, however in lower airway they were enriched with additional T cell specificities (Suppl. Figure 5A). In the case of SARS-CoV-2 CD8⁺ T cells, their antigen recognition profile was more diverse in both sites, with S-specific CD8⁺ T cells being less dominant (Suppl. Figure 5B).

**Longevity of antibody and T cells responses in lung mucosa following infection and vaccination or vaccination alone**

In an attempt to assess the longevity of vaccine-induced SARS-CoV-2 immune memory in the lung mucosa following vaccination alone or in combination with infection (primary or breakthrough infection), antibody and T cell responses assessed in BAL and paired blood of

**Fig. 2 | Detectable anti-SARS-CoV-2 memory B cell responses in the lung mucosa following infection and vaccination. A** Example flow cytometry plots of Spike-, RBD- and N-specific global memory B cells (MBCs) in PBMC and BAL sample of an unexposed pre-pandemic control (left) and an infected vaccinated donor(-right) (see Suppl. Fig. 3 for full gating). **B**–**D** Frequency of circulating Spike-, RBD- and N-specific MBCs in control ($n = 10$), uninfected vaccinated ($n = 9$) and infected vaccinated donors ($n = 22$). **E**–**G** Frequency of Spike-, RBD- and N-specific MBCs detected in BAL samples of control ($n = 6$), uninfected vaccinated ($n = 5$) and infected vaccinated donors ($n = 13$). The limit of assay sensitivity (LOS) per antigen is depicted with dotted black line. **H** Frequency of SARS-CoV-2 specific memory B cells in blood (PBMC) and BAL, shown as paired samples, of uninfected vaccinated ($n = 5$) and infected vaccinated donors ($n = 13$). **I** Distribution of global MBC subsets in PBMC and BAL based on the expression of IgD and IgM in uninfected vaccinated and infected, vaccinated donors together ($n = 18$). Homologous vaccination with ChAdOX1_S or mRNA vaccine is depicted with an open or close circle, respectively and heterologous vaccination with semi-full circle. Data are presented as median values and interquartile ranges (IQRs). Statistical differences were determined by Kruskal–Wallis test following correction for multiple comparisons (**B**–**D**), two-sided Mann–Whitney $U$ test (**E**–**G**) and two-sided Wilcoxon's paired test (**H**, **I**). Source data are provided as a Source Data file.

uninfected and infected vaccinated individuals were plotted in association with time post the last 'immunising event' (last dose of vaccination or breakthrough infection) from sample collection. These temporal correlations should be interpreted with caution given the small cross-sectional cohort studied in the present work.

Levels of circulating anti-S and anti-RBD IgG were negatively correlated with time post-vaccination in the infection-naïve but not the infected vaccinated group, implying quicker antibody decay in the former (Fig. 5A). In the lung mucosa, anti-S and anti-RBD IgG titres suggested a fast rate of decay in the infected vaccinated group, although remained detectable at 7 months post last exposure to viral spike (Fig. 5B). On the other hand, levels of anti-S and RBD IgA in BAL, detectable following infection, quickly reached pre-pandemic levels (at 150 and 90 days, retrospectively), Fig. 4C. This result is in agreement with previous studies in convalescent patients that reported short-lived IgA-mediated immunity at mucosal sites[22,45].

Circulating and lower-airway S-specific T cell frequencies were also plotted in association with time post the last immunising event (latest vaccination or infection) in both vaccinated groups. In blood, the levels of S-specific CD4+ and CD8+ T cells in the infected vaccinated group reached those induced by vaccination alone at approaching 7 months post-last immunising event (Fig. 6A). In the lung mucosa, cross-sectional sample analysis suggested that S-specific T cell responses may be better preserved locally following hybrid immunity. Despite a trend of decay over time post last exposure to spike, S-specific CD4+ T cell responses were detectable in the lung mucosa of infected vaccinated individuals up to 7-months after the last immunising event. Lower-airway S-specific CD8+ T cells did not associate negatively with time and followed a relatively stable trajectory throughout the period of 7-months (Fig. 6B, C). In a few cases, the human lung mucosa retained partial immune memory to SARS-CoV-2 for several months after infection. T cell specificities not affected by SARS-CoV-2 vaccination, such as M-, N- and RTC-specific CD4+ and CD8+ T cells, were detectable at variable frequencies between 3 to 19 months post infection, showing a tendency to be less frequent after longer intervals since exposure (Suppl. Fig. 5C, D).

## Discussion

Despite high SARS-CoV-2 seroprevalence globally by either vaccination or infection, regular waves continue to cause breakthrough infections. Immunological memory to SARS-CoV-2 in the respiratory mucosa following vaccination, infection and hybrid immunity is not well understood. Even though immune responses to SARS-CoV-2 after infection and vaccination have been extensively studied in blood, there is an ongoing controversy regarding the magnitude of T cells responses in the context of hybrid immunity[46]. Protective mucosal immunity could be harnessed for the development of vaccines specifically targeting protection against airway infection to block transmission of the virus in the population.

Here, we assessed the potential of peripheral SARS-CoV-2 vaccination to induce anti-viral immune memory in the human lung mucosa and whether infection would influence the immunological outcome. We found that following SARS-CoV-2 vaccination alone, the airway mucosa contained detectable spike IgG, but levels were increased and accompanied by mucosal IgA in infected vacinees. Spike-specific memory B cells were only detectable in BAL in donors with history of infection and vaccination. Importantly, parenteral SARS-CoV-2 vaccination (mRNA or adenoviral vector vaccine) did not appear to seed the human respiratory mucosa with tissue-resident spike-specific T cells, despite the induction of notable T cell responses in the circulation. Compared to SARS-CoV-2 vaccination alone, infection and vaccination resulted in higher humoral and cellular immune responses against the vaccine antigens in the periphery. In contrast to vaccination alone, infection and vaccination generated a heightened and potentially persistent spike-specific T cell reservoir in the human lung mucosa, complemented in some cases with local memory MBC and T cell responses against additional SARS-CoV-2 antigens. A long-lived, airway-compartmentalised B and T cell reservoir in the lung mucosa may confer better recognition of Omicron sublineages and future variants and protect against severe disease, supporting the need for vaccines specifically targeting the airways.

In line with others[17,18,32], our results indicate that following vaccination alone, SARS-CoV-2 immunity in the respiratory mucosa is limited to humoral immunity, with IgG dominating over IgA titres against the vaccine-antigens. Induction of both anti-Spike IgG and IgA was more efficient in the lung mucosa of vaccinated individuals with prior infection. The strong correlation observed between anti-S IgG in serum and BAL samples supports the notion that systemic antibodies elicited by vaccination transudate to the respiratory mucosa, as previous vaccination studies have reported[47,48]. Despite the key role of antibodies in neutralising the virus at the respiratory mucosa- the primary site of infection, local humoral immunity wanes quickly[22] making individuals more susceptible to immune escape by Omicron sublineages and future variants[17]. In addition, findings from other respiratory infection and vaccination studies indicate that higher levels of antibodies are required in the nasal mucosa to protect against local infection compared to levels required in blood to protect against invasive disease[49]. However, the finding of class-switched memory B cells enriched in the lung mucosa raises the possibility they produce a repertoire of antibodies better able to cross-recognise variants, as shown in other infections[4]. Mucosal antibodies may also harness local lung cells such as NK cells and phagocytes for non-neutralising Fc-dependent cellular immunity.

Booster parenteral vaccination is required to enhance waning humoral immunity, but the frequency and intensity of robust systemic T-cell responses is not boosted by additional vaccination[50]. With respect to peripheral immune responses, our findings were in agreement with studies reporting that hybrid immunity elicits considerably high humoral and cellular responses than immune responses induced solely by vaccination[33,51,52]. Studying the human lung mucosa, we provide the first evidence, to our knowledge, that infection and vaccination, contrary to SARS-CoV-2 immunisation alone, can generate broad, and potentially long-lived anti-viral immune responses in the lower airways. In line with other studies[17,19], the intensity of S-specific B and T cell responses were enriched in the lung mucosa compared to the periphery and similarly, enriched B and T cells responses were detected for other SARS-CoV-2 antigens. Our limited cross-sectional data suggested athe potential for lower-airway localised B and T cells

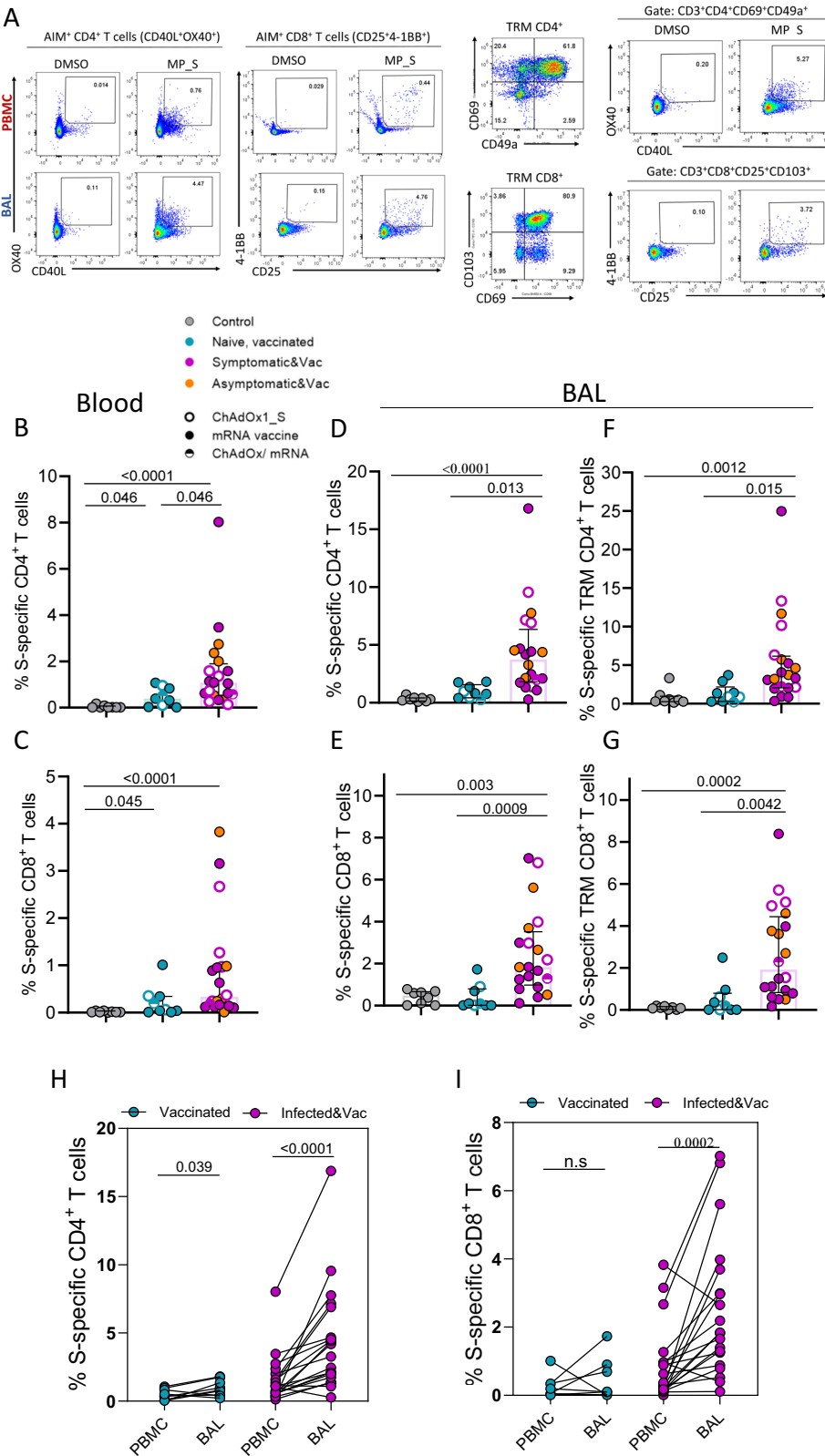

reservoirs to be long-lived, however larger studies are required that are powered to address this aspect of local immunity. In studies of other respiratory viruses, lung localised, tissue-residing B and T cells associate with protection in mouse models of influenza[5,53] and RSV infection[10]. In human challenge models of influenza and RSV infection, enrichment of CD4[+] and CD8[+] $T_{RM}$ cells in the airways was associated with mitigated respiratory symptoms, viral control, and reduced disease severity[12,54]. The prolonged memory together with the ability of T cells to better recognise more conserved parts of SARS-CoV-2, support the utility of developing multi-specific mucosally-administered vaccines that could boost tissue localised and resident memory T and B cells in the lung mucosa. Preclinical studies of SARS-CoV-2 have demonstrated that intranasal vaccination decreases viral shedding and transmission relative to parenteral vaccines[28,55,56]. In addition, the

**Fig. 3 | Detection of Spike-specific T cells responses in the lung mucosa after infection and vaccination but not following vaccination alone.**
**A** Representative flow cytometry plots of S-specific CD4+ and CD8+ T cells in PBMC and BAL sample (on the left) and S-specific tissue-resident memory (TRM) CD4+ and CD8+ T cells in BAL sample (on the right) of an infected, vaccinated donor. Identification of S-specific T cells was based on the AIM assay, assessing co-expression CD40L and OX40 on CD4+ T cells and co-expression of CD25 and 4-1BB on CD8+ T cells after stimulation with Spike megapool. **B**, **C** Frequency of circulating S-specific CD4+ and CD8+ T cells in control (*n* = 8), uninfected vaccinated (*n* = 9) and infected, vaccinated donors (*n* = 20). **D**−**G** Frequency of lower-airway S-specific CD4+ and CD8+ T cells within the global (**D**, **E**) and TRM compartment (**F**, **G**) in control (*n* = 8), naïve, vaccinated (*n* = 9) and infected, vaccinated donors (*n* = 20). **H**, **I** Frequency of S-specific CD4+ and CD8+ T cells in PBMC and BAL, shown as paired samples, of uninfected vaccinated (*n* = 9), and infected vaccinated donors (*n* = 20). Homologous vaccination with ChAdOx1_S or mRNA vaccine is depicted with an open or close circle, respectively and heterologous vaccination with semi-full circle. Data are presented as median values and interquartile ranges (IQRs). Statistical differences were determined by Kruskal−Wallis test following correction for multiple comparisons (**B**−**G**) and two-sided Wilcoxon's paired test (**H**, **I**). Source data are provided as a Source Data file.

combined approach of systemic priming SARS-CoV-2 vaccination followed by intranasal boosting with either adenovirus vectored vaccines or an adjuvanted Spike vaccine elicited both systemic and protective mucosal immunity with cross-reactive properties[57,58]. Yet in humans, there are limited data on the immunogenicity of SARS-CoV-2 vaccines that target the airways, focusing mainly on humoral immunity[59], and an ongoing controversy as to whether peripheral vaccination can induce mucosal responses[17,18,30,31] a critical point to inform the need for mucosal-targeted vaccines.

Our study is not without limitations. Due to increased hesitancy towards the bronchoscopy procedure throughout the study and the time-sensitive setting, we were not able to recruit a larger cohort of study participants. The fast roll out and uptake of the COVID-19 vaccine programme in the UK prevented the inclusion of convalescent unvaccinated individuals which would allow us to compare hybrid airway immunity with that induced by infection alone. The low BAL cell yields restricted the analysis of other T cell specificities to selected SARS-CoV-2 proteins and only allowed the assessment of vaccine-induced memory B cells in the lung mucosa in a subset of vaccinated individuals. In addition, the study was not powered to accurately describe the kinetics of humoral and cellular responses over time after exposure to the virus or vaccination, hence temporal associations (or lack of them) must be interpreted with caution. Although, we were able to detect and characterise T cell specificities for over a year post infection and up to 7 months post the last exposure to spike, our study was not designed to describe the longevity of lower-airway localised B and T cells reservoirs. Therefore, future longitudinal and statistically powered studies are needed to fully understand the long-term impact of airway localised T and B cell immunity in SARS-CoV-2 protection.

Overall, our data suggest airway mucosal B and T cell immunity against SARS-CoV-2 is enhanced following infection and vaccination, as opposed to peripheral vaccination alone. Vaccines that induce airway localised memory T and B cells may provide broader long-term protection at the site of infection to allow more efficient shut-down of infection and reduced onward virus transmission.

## Methods
### Study design and cohorts
This was a cross-sectional study, which included a cohort of SARS-CoV-2 vaccinated individuals (*n* = 31), who had received two or three doses of mRNA or the ChAdOx1_S adenoviral vector vaccine or a combination of those (details shown on Table 1). A subset of them had experienced PCR-confirmed symptomatic infection (*n* = 17) or serologically confirmed asymptomatic infection (*n* = 5), referred to the group of infected and vaccinated individuals (*n* = 22), whereas the remaining vaccinees had not experienced SARS-CoV-2 infection (uninfected vaccinated, *n* = 9). BAL samples were obtained through research bronchoscopy 1 to 13 months (23-392 days) after the last vaccine dose and 1 to 19 months (37−570 days) after symptoms onset for those who had experienced SARS-CoV-2 infection. The infected individuals were either had been admitted to hospital between April 2020 to January 2021 (*n* = 12), when the ancestral SARS-CoV-2 strain was still dominant in the UK or had experienced a mild infection between April 2022 to November. Blood samples for sera and PBMC isolation were collected

at the same day as BAL. Pre-pandemic samples from healthy, unexposed individuals (*n* = 11), collected from 2015 to 2018, were also included into the analysis, as a control group (Fig. 1A). The demographic and clinical characteristics of the 3 study groups are shown in Table 1 and study's recruitment process is presented in a consort flow chart (Supplemental Fig. 1).

### Sample processing
BAL samples were processed as previously described[60], cryopreserved in CTL-CryoABC medium kit (Immunospot). After thawing, alveolar macrophages were routinely separated from other non-adherent immune cell populations using an adherence step, as previously described[61]. Blood was processed for sera collection or PBMCs were isolated from heparinized blood samples using density-gradient sedimentation layered over Ficoll-Paque in SepMate tube and then cryopreserved in CTL-CryoABC medium kit (Immunospot).

### ELISA for SARS-CoV-2 proteins
ELISA was used to quantify levels of IgG and IgA against Spike trimer, RBD and N in serum and BAL samples, as previously described[62]. Briefly, 96-well plates (U bottom) were coated with 1 µg/ml SARS-CoV-2 antigen and stored at 4°C overnight for at least 16 h. The next day, plates were washed 3 times with PBS/0.05% Tween-20 and blocked with 2% BSA or 1% casein in PBS for 1 h at room temperature. Sera and BAL diluted in 0.1% BSA-PBS were plated in duplicate and incubated for 2 h at room temperature alongside an internal positive control (dilution of a convalescent serum) to measure plate to plate variation. For the standard curve, a pooled sera of SARS-CoV-2 infected participants was used in a two-fold serial dilution to produce either eight or nine standard points (depending on the antigen) that were assigned as arbitrary units. Goat anti-human IgG (γ-chain specific, A9544, Millipore-Sigma) or IgA (α-chain specific, A9669, Millipore-Sigma or 2050-04, Southern Biotech) conjugated to alkaline phosphatase was used as secondary antibodies, and plates were developed by adding 4-nitrophenyl phosphate in diethanolamine substrate buffer. Optical densities were measured using an Omega microplate reader at 405 nm. Blank-corrected samples and standard values were plotted using the 4-Parameter logistic model (Gen5 v3.09, BioTek).

### B cells immunophenotyping and detection of SARS-CoV-2 specific B cells
Cryopreserved BAL cells and PBMCs were used for detection of SARS-CoV-2 specific B cells in lower airways and blood, respectively. Biotinylated tetrameric S, RBD and N protein were individually labelled with different streptavidin conjugates at 4°C for 1h[62]. Biotinylated S and RBD were directly labelled with Streptavidin-PE (with a ratio 1:3 and 1:5.7, respectively); with Streptavidin-BV570 (S with a ratio 1:2.7); and Streptavidin-BV785 (RBD with a ratio 1:5). Biotinylated N protein was labelled with Streptavidin-PE (with a ratio of 1:2.3) and Streptavidin-AF647 (N protein with a ratio 1:0.5).

PBMCs and BAL cells were thawed and stained with Live/dead e506 viability dye and an antibody cocktail for surface markers for 30 min in the dark, washed twice and resuspended in 200 µL of PBS. Parallel samples stained with an identical panel of monoclonal

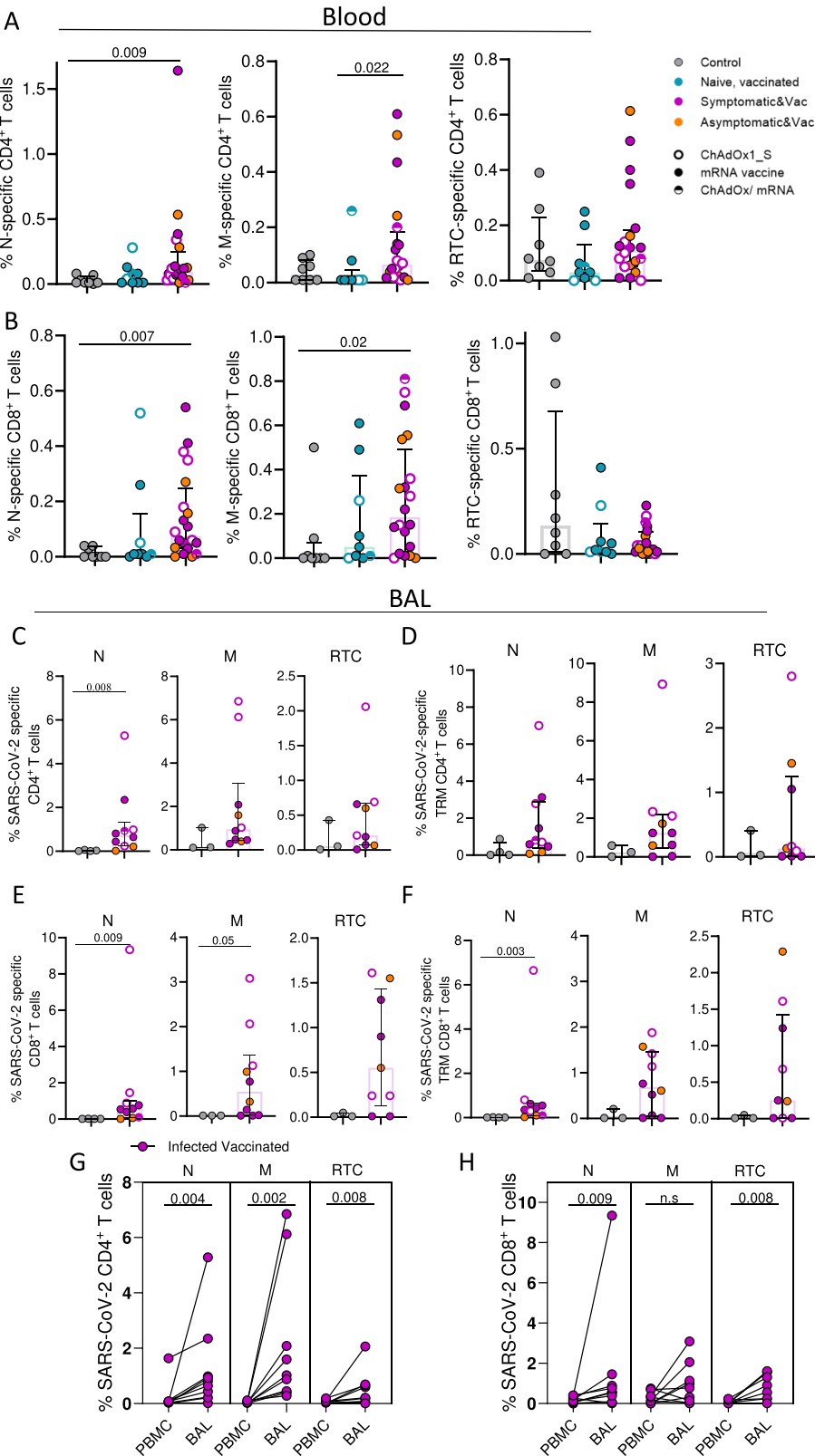

**Fig. 4 | Detection of infection-induced SARS-CoV-2 T cell responses in the periphery and lung mucosa. A**, **B** Frequency of circulating N-, M- and RTC-specific CD4+ and CD8+ T cells in control (n = 8), uninfected, vaccinated (n = 9) and infected, vaccinated donors (n = 20). **C**–**F** Frequency of N-, M- and RTC-specific CD4+ and CD8+ T cells within the global and tissue-resident memory (TRM) compartment of lower airway T cells in control (n = 4) and infected vaccinated donors (up to n = 10). **G**, **H** Frequency of SARS-CoV-2-specific CD4+ and CD8+ T cells in PBMC and BAL, shown as paired samples, of infected vaccinated donors (n = 10 for N and M and

n = 9 for RTC). Homologous vaccination with ChAdOX1_S or mRNA vaccine is depicted with an open or close circle, respectively and heterologous vaccination with semi-full circle. Data are presented as median values and interquartile ranges (IQRs). Statistical differences were determined by Kruskal–Wallis test following correction for multiple comparisons (**A**, **B**), two-sided Mann–Whitney U test (**C**–**F**) and two-sided Wilcoxon's paired test (**G**, **H**). Source data are provided as a Source Data file.

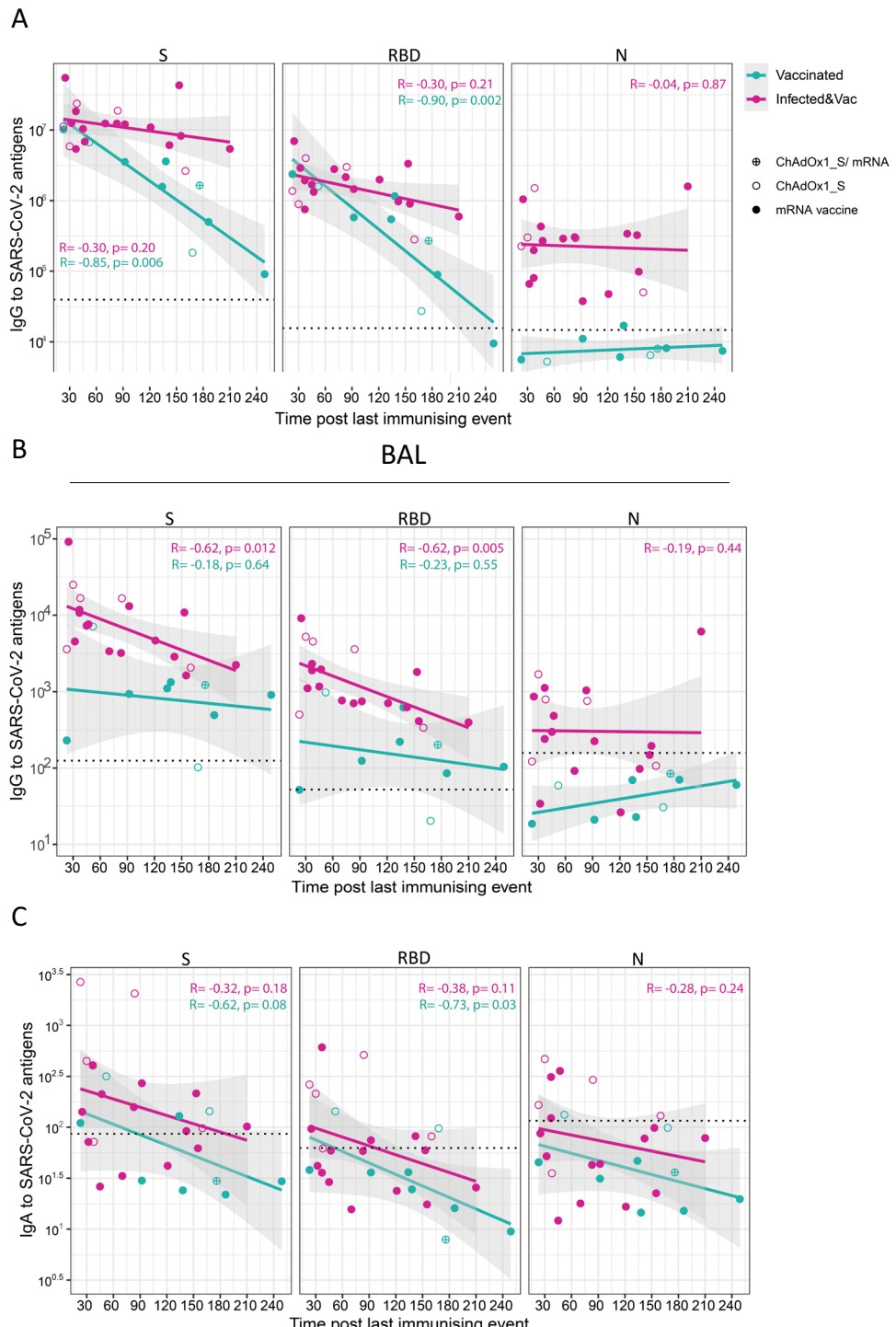

**Fig. 5 | Kinetics of antibody responses to SARS-CoV-2 antigens in blood and the lung mucosa following vaccination and infection. A, B** Correlation between time post-vaccination and levels of IgG to S, RBD and N proteins measured in serum (**A**) and BAL supernatant (**B**) of uninfected vaccinated ($n = 9$) and infected vaccinated individuals ($n = 22$). **C** Correlation between time post-vaccination and levels of IgA to S, RBD and N proteins measured in BAL supernatant of uninfected, vaccinated ($n = 9$) and infected, vaccinated individuals ($n = 22$). The limit of assay sensitivity (LOS) per antigen is depicted with dotted black line. Homologous vaccination with ChAdOX1_S or mRNA vaccine is depicted with an open or close circle, respectively. Results of two-tailed Spearman correlation test and linear regression line with 95% confidence interval (grey shading) are shown. Source data are provided as a Source Data file.

antibodies (mAbs) but excluding the SARS-CoV-2 proteins (fluorescence minus one [FMO] controls), were used as controls for non-specific binding. All samples were acquired on an Aurora cytometer (Cytek Biosciences) and analysed with Flowjo software version 10 (Treestar). The flow-cytometry panel of mAbs used to phenotype

global and antigen-specific subsets can be seen in Supplementary Table 1.

The frequency of antigen-specific B cells was calculated within the fraction of MBCs (CD19$^+$CD27$^+$, excluding the naïve IgD$^+$CD27$^-$ and the double negative IgG$^-$CD27$^-$ fractions, see gating strategy

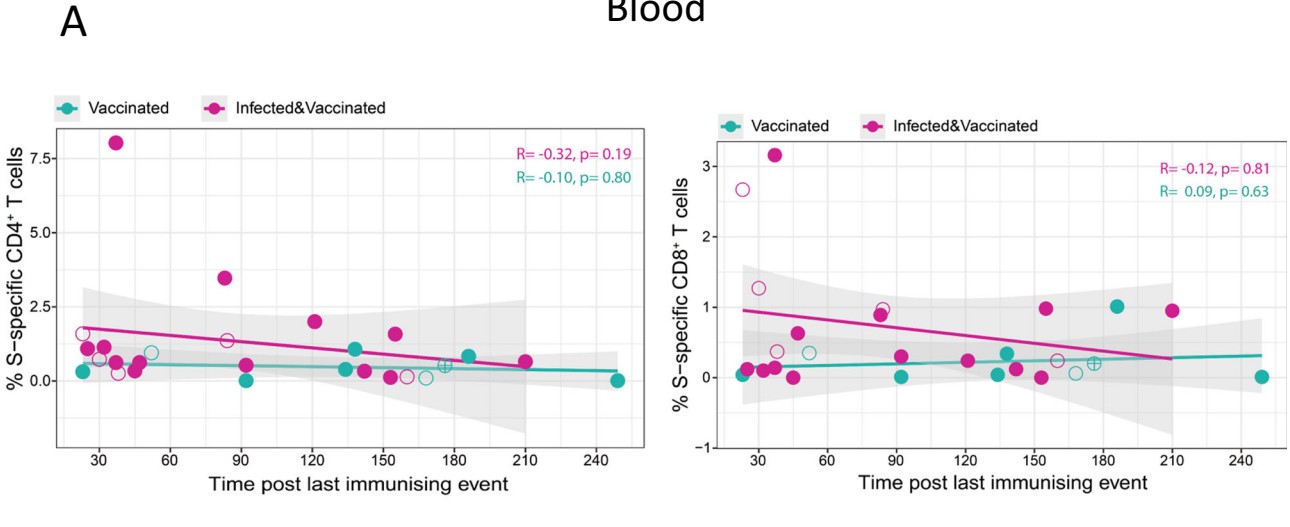

## Blood

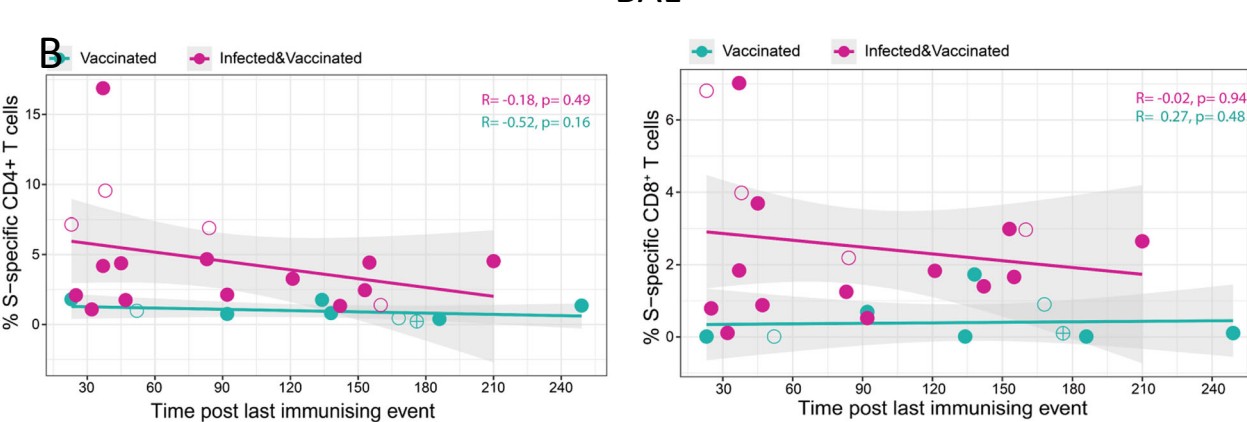

## BAL

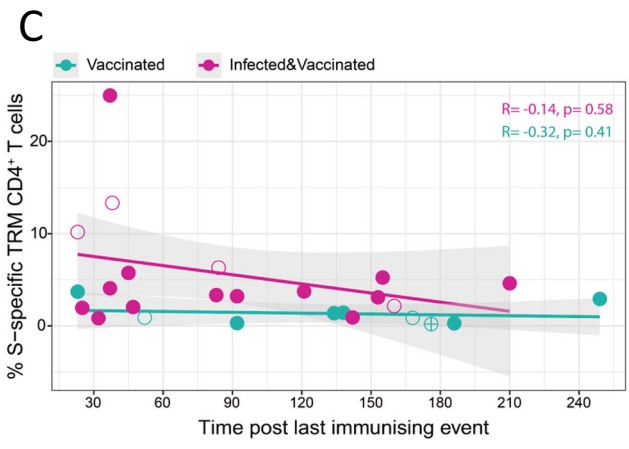

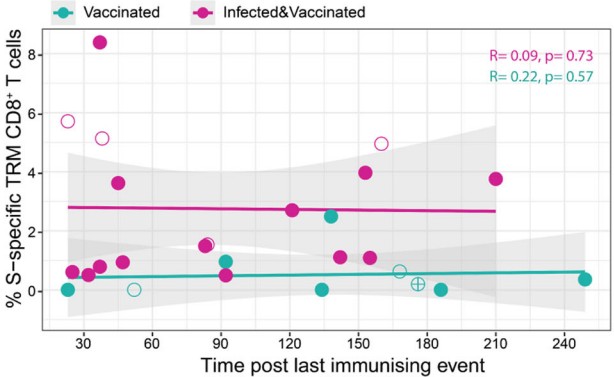

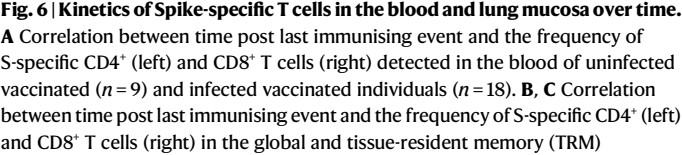

**Fig. 6 | Kinetics of Spike-specific T cells in the blood and lung mucosa over time.**
**A** Correlation between time post last immunising event and the frequency of
S-specific CD4⁺ (left) and CD8⁺ T cells (right) detected in the blood of uninfected
vaccinated ($n = 9$) and infected vaccinated individuals ($n = 18$). **B, C** Correlation
between time post last immunising event and the frequency of S-specific CD4⁺ (left)
and CD8⁺ T cells (right) in the global and tissue-resident memory (TRM)
compartment of lower-airway T cells in BAL of uninfected vaccinated ($n = 9$) and
infected vaccinated individuals ($n = 18$). Homologous vaccination with ChAdOX1_S
or mRNA vaccine is depicted with an open or close circle, respectively and het-
erologous vaccination with semi-full circle. Results of two-tailed Spearman corre-
lation test and linear regression line with 95% confidence interval (grey shading)
are shown.

(Suppl. Figure 3A). For phenotypic analysis of spike-, RBD-, and N-specific B cells, a sufficient magnitude of responses (≥50 cells in the relevant parent gate) was required.

## Activation-induced markers (AIM) T cell assay

Mononuclear BAL cells ($1 \times 10^5$ cells per well) and PBMCs ($1 \times 10^6$ cells per well) were seeded in 96-well plates in RPMI supplemented with 1% PNS and 10% AB human serum (Merck, UK) and stimulated with SARS-CoV-2 specific peptides pools. The peptides pools used were spanning the whole Spike protein (15-mer peptides overlapping by 10 amino acids)[63] or overlapping peptides spanning the immunogenic domains of the SARS-CoV-2 N (Prot_N) and M (Prot_M) purchased from Miltenyi Biotec[62] or combined pools spanning SARS-CoV-2 NSP7, NSP12 and NSP13 proteins (15-mer peptides overlapping by 10 amino acids) of the ancestral SARS-CoV-2 strain[14,64]. Prior to the peptide addition, cells were blocked with 0.5 µg/ml of anti-CD40 mAb (Miltenyi Biotec) for 15 min at 37°C. A stimulation with an equimolar amount of DMSO was performed as a negative control and Staphylococcal enterotoxin B (SEB, 2 µg/mL) was included as a positive control. The following day cells were harvested from plates, washed and stained for surface markers (Supplemental table 2 and 3).

AIM+ CD4+ T cells were identified as CD40L+OX40+, 4-1BB+OX40+, 4-1BB+CD40L+ subsets, and the CD40L+OX40+ combination was used to quantify SARS-CoV-2 specific CD4+ T cells frequency. SARS-CoV-2 specific CD8+ T cells were identified as 4-1BB+CD25+. Antigen-specific CD4+ and CD8+ T cells were measured and presented as DMSO background−subtracted data.

## Statistical analysis

Participant characteristics were summarised as n, median (interquartile range) or frequency (percentage). Chi-squared test and Fisher's exact test were conducted to identify any significant changes in categorical variables. Non-parametric Wilcoxon paired tests and Mann-Whitney tests were conducted to compare quantitative data within the same group or between two groups, respectively. In addition, Kruskal−Wallis rank-sum test with Dunn's correction were performed to compare quantitative data amongst groups (three groups comparison). Tests were two-sided with an α level of 0.05. For correlations, two-tailed Pearson's or Spearman's $r$ test was used. To explore the association between time after infection and vaccination, we employed a linear regression model. Data were analysed in R software version 4.0.3 (R Foundation for Statistical Computing, Vienna, Austria), using rstatix package (version 4.2.3) or in Graphpad Prism version 9.0.

## Ethics statement

All volunteers gave written informed consent and research was conducted in compliance with all relevant ethical regulations. Ethical approval was given by the NorthWest National Health Service Research Ethics Committee (REC no. 18/NW/0481 and Human Tissue licensing no. 12548). All participants provided written informed consent and were free to withdraw from the study at any point.

## Reporting summary

Further information on research design is available in the Nature Portfolio Reporting Summary linked to this article.

## Data availability

All data generated and analysed during the present study are included in this published article and its supplementary information files. A Source Data file is provided with this paper. All relevant data are also available from the authors. Source data are provided with this paper.

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

## Acknowledgements

We thank all the patients and healthy volunteers who participated in the present study and all the clinical staff who helped with recruitment and sample collection. The study was financially supported by EPSRC grant (no. EP/W016389/1) awarded to E.M and D.M.F. Collection of pre-pandemic clinical samples was supported by the Bill and Melinda Gates Foundation (grant no. OPP1117728) and the UK Medical Research Council (grant no. M011569/1) awarded to D.M.F. The

production of peptide pools has been funded or in part with federal funds from the National Institutes of Health, Contract No. 75N9301900065 awarded to A.S and D.W. A Wellcome Trust Investigator Award (no. 214191/Z/18/Z) and CRUK Immunology grant (no.26603) have been awarded to M.K.M.

## Author contributions

E.M., M.O.D, M.K.M. and D.M.F. conceived and designed the study. A.H.W. and M.F. recruited and consented study participants. A.H.W., M.F., R.R., K.L. and A.M.C. obtained human samples. E.M., J.R., J.H. and C.S. processed samples. E.M., M.O.D., J.R., J.H., O.O., S.B.R., E.S. and T.L. generated and analysed the data. E.M., M.O.D., M.K.M. and D.M.F interpreted data. E.M., J.R. and B.U. developed the assays. S.J.D., D.W. and A.S. provided material for the assays. E.M. and M.O.D. wrote the manuscript and with subsequent inputs from the co-authors. All co-authors approved the final version of the manuscript.

## Competing interests

A.S. is a consultant for Gritstone Bio, Flow Pharma, Moderna, AstraZeneca, Qiagen, Fortress, Gilead, Sanofi, Merck, RiverVest, MedaCorp, Turnstone, NA Vaccine Institute, Emervax, Gerson Lehrman Group and Guggenheim. La Jolla Institute has filed for patent protection for various aspects of T cell epitope and vaccine design work. The remaining authors declare no competing interests.
