## [Peer Review File · Nature Communications]

Respiratory mucosal immune memory to SARS-CoV-2 after infection and vaccinationEditorial Note: This manuscript has been previously reviewed at another journal that is not operating a transparent peer review scheme. This document only contains reviewer comments and rebuttal letters for versions considered at *Nature Communications*.

REVIEWERS' COMMENTS

Reviewer #1 (Remarks to the Author):

A strength of this study is the paired analysis of PBMC & BAL in uninfected vaccinated and infected vaccinated individuals. A limitation of the study is that the numbers studied are small. Nevertheless, there are several key noteworthy results in this manuscript. For example, the fact that there are enriched T and B cell responses in the BAL compared to the periphery in the infected vaccinated individuals compared to vaccinated only and an enriched class switched MBC population in the lower human airways.

The work is of significance to the field and important. It to some extent re-reports published findings, but also adds several original observations.

The data as presented does not always appear to support fully the big conclusions and claims being made and some of the conclusions about longevity of T cell responses need to be toned down somewhat. With this in mind, the last two sentences of the abstract could be edited as follows:

'Spike-specific T cells persisted in the lung mucosa for 7 months after the last immunising event. Thus, peripheral vaccination alone does not appear to induce durable lung mucosal immunity against SARS-CoV-2, supporting an argument for the urgent need for vaccines targeting the airways.'

Extended Data Figure 3E, F is a little confusing as it is currently presented. It shows the correlation between time post infection from symptoms onset and the frequency of N, M and RTC specific CD4 & CD8 T cells detected in the BAL of infected vaccinated individuals (n=15). Please can Extended Data Figure 3E, F be annotated with the actual number of individuals tested and the number of peptide panels tested per individual at each time point post infection.

The methodology is sound and meets the expected standards in the field.

There is enough detail provided in the methods for the work to be reproduced.

REVIEWERS' COMMENTS

Reviewer #1 (Remarks to the Author):

A strength of this study is the paired analysis of PBMC & BAL in uninfected vaccinated and infected vaccinated individuals. A limitation of the study is that the numbers studied are small. Nevertheless, there are several key noteworthy results in this manuscript. For example, the fact that there are enriched T and B cell responses in the BAL compared to the periphery in the infected vaccinated individuals compared to vaccinated only and an enriched class switched MBC population in the lower human airways.

The work is of significance to the field and important. It to some extent re-reports published findings, but also adds several original observations.

We thank the reviewer for acknowledging the importance and significance of our work to the field, as well as the presence of several original observations.

The data as presented does not always appear to support fully the big conclusions and claims being made and some of the conclusions about longevity of T cell responses need to be toned down somewhat. With this in mind, the last two sentences of the abstract could be edited as follows:

'Spike-specific T cells persisted in the lung mucosa for 7 months after the last immunising event. Thus, peripheral vaccination alone does not appear to induce durable lung mucosal immunity against SARS-CoV-2, supporting an argument for the urgent need for vaccines targeting the airways.'

We thank the Reviewer for the suggestion. These sentences appear now as above in the abstract. We have also toned down conclusions drawn from underpowered datasets, particularly claims on T cell responses longevity and we have further discussed in the study limitations this exploratory observation.

Extended Data Figure 3E, F is a little confusing as it is currently presented. It shows the correlation between time post infection from symptoms onset and the frequency of N, M and RTC specific CD4 & CD8 T cells detected in the BAL of infected vaccinated individuals (n=15). Please can Extended Data Figure 3E, F be annotated with the actual number of individuals tested and the number of peptide panels tested per individual at each time point post infection.

Extended Data Figure 3E, F have now been annotated accordingly to capture the requested information.

The methodology is sound and meets the expected standards in the field. There is enough detail provided in the methods for the work to be reproduced.

We thank the Reviewer for appreciating the quality and robustness of our methodology.